# Polypharmacy occurrence and the related risk of premature death among older adults in Denmark: A nationwide register-based cohort study

Anna Vera Jørring Pallesen[1,2]*, Maria Kristiansen[2,3], Rudi G. J. Westendorp[1,2,3], Laust Hvas Mortensen[1,2]

1 Methods and Analysis, Statistics Denmark, Copenhagen, Denmark, 2 Department of Public Health, University of Copenhagen, Copenhagen, Denmark, 3 Center for Health Aging, University of Copenhagen, Copenhagen, Denmark

* x30@dst.dk

**Data Availability Statement:** Data cannot be shared publicly because individual level data is not allowed to be shared. Data are available from the Statistics Denmark Institutional Data Access for

## Abstract

### Background

Polypharmacy, defined as the concurrent use of ≥5 medications, increases the risk of drug-drug and drug-disease interactions as well as non-adherence to drug therapy. This may have negative health consequences particularly among older adults due to age-related pharmacokinetic and pharmacodynamic changes. This study aims to uncover the occurrence of polypharmacy among older adults in Denmark and investigate how polypharmacy relates to mortality.

### Method

This nationwide register-based study included 1,338,058 adults aged 65+ years between January 2013 and December 2017 in Denmark. Polypharmacy prevalence was measured at time of inclusion while incidence and the association between polypharmacy and mortality were measured over the five-year follow-up using Cox regression. In an attempt to adjust for confounding by indication, propensity scores with overlap weighting were introduced to the regression model.

### Results

At time of inclusion, polypharmacy prevalence was 29% and over the five years follow-up, 47% of the remaining adults transitioned into polypharmacy. Identified risk factors included multimorbidity (2+ morbidities: HR = 3.51; 95% CI = 3.48–3.53), age (95+ years: HR = 2.85; 95% CI = 2.74–2.96), socioeconomic factors (Highest income quartile: HR = 0.81; 95% CI = 0.80–0.81), region of birth region (Non-western migrants: HR = 0.77; 95% CI = 0.75–0.79), marital status (Divorced: HR = 1.10; 95% CI = 1.10–1.12) and year of inclusion (2017: HR = 1.19; 95% CI = 1.19–1.22). Further analyses showed that polypharmacy involves many different drug cocktails with medication for the cardiovascular system (95%), blood and

researchers who meet the criteria for access to confidential data (contact via forskningsservice@dst.dk).

**Funding:** AVJP, LHM and RGJW was supported by grants from Novo Nordisk Foundation [NNF17OC0027812] (https://novonordiskfonden.dk/en/), while MK was funded by Nordea Fonden [02-2017-1749] (https://nordeafonden.dk/). The funders had no role in study design, data collection and analysis, decision to publish, or preparation of the manuscript.

**Competing interests:** The authors have declared that no competing interests exist.

blood-forming organs (69%), alimentary tract and metabolism (61%) and nervous system (54%) contributing the most. After adjustment for propensity scores with OW, both polypharmacy (HR = 3.48, CI95% = 3.41–3.54) and excessive polypharmacy (HR = 3.48, CI95% = 3.43–3.53) increased the risk of death substantially.

## Conclusion

A considerable proportion of older adults in Denmark were exposed to polypharmacy dependent on health status, socio-economic status, and societal factors. The associated three- to four-fold mortality risk indicate a need for further exploration of the appropriateness of polypharmacy among older adults.

## Background

The proportion of older adults, defined as adults aged 65 or more years, in the population is increasing, and it is predicted to continue as life expectancy continues to increase [1]. Older adults are at increased risk of developing chronic conditions for which drug therapy is often the first choice of treatment [2, 3]. Chronic conditions are most often treated according to single disease guidelines, which in particular among older adults with multiple chronic conditions may result in complex drug regimens. Also, older adults with multiple chronic conditions often experience that several different health care professionals are involved with their treatments. In these cases, communication between professionals is highly important in order to ensure quality in healthcare delivery, including appropriate medication prescription [4]. New medications and expansion of treatment at higher ages are also likely to fuel the number of older adults receiving drug therapy. Consequently, the intake of many medications is common among older adults. A nationwide study conducted in Sweden found that adults aged 65 years or older had a mean intake of 4.6 different medications [5]. A nationwide Danish study found that 50% of adults aged 60 years were exposed to polypharmacy defined as an intake of at least five medications [6]. Also, several studies show a marked increase in excessive polypharmacy, which is defined as a concurrent intake of 10 or more different medications [7–9]. Polypharmacy can be clinically appropriate if the medications improve health as intended. However, polypharmacy is an important challenge for clinicians, as many older adults are exposed to polypharmacy to the extent that drug therapy is no longer beneficial [5].

Several studies have investigated the health of older adults with polypharmacy and found increased risks of hospitalisation, falls, frailty, lowered cognitive functions, and mortality [10–12]. Poor health outcomes in those with polypharmacy may be caused by morbidities being the primary indication for medication use. However, there is reason to believe that polypharmacy in and of itself may have adverse health consequences. Firstly, using single disease clinical guidelines to treat older adults with multiple chronic conditions creates the risk of potentially harmful drug-disease interactions [13]. With multiple chronic conditions, the body may react differently to medications than anticipated, while drugs intended for the treatment of one condition may have harmful effects on another condition. Secondly, the risk of drug-drug interactions increases exponentially with the number of drugs consumed [10]. Studies have identified a wide range of harmful drug-drug interactions, however, with a high number of drugs, the complexity increases, and multiple drugs may interact in unknown ways [14, 15]. Thirdly, older adults are particularly prone to adverse drug events due to changes in pharmacokinetics and pharmacodynamics at older ages. These changes make older adults more

sensitive to the effect of drugs; moreover, older adults may be more sensitive to drug-drug and drug-disease interactions.

Given these concerns, this study examines the prevalence and incidence of polypharmacy as well as characteristics associated with an increased risk of polypharmacy. In particular, this study intends to identify the role of age and period effects. Also, we wish to examine how polypharmacy relates to mortality. Lastly, we investigate which medications are most often taken in combination among polypharmacy-exposed older adults.

## Methods and material

### Study design and population

We used an open cohort for this study consisting of all individuals aged 65 years or more and living in Denmark at some point between 1 January 2013 and 31 December 2017 (N = 1,338,058). Exposure to polypharmacy and excessive polypharmacy in the study population was investigated in two designs: 1) a cross-sectional study investigating the prevalence of polypharmacy at time of inclusion, and 2) a longitudinal cohort study investigating the incidence of polypharmacy over a five-year period among the population that were not exposed to polypharmacy at time of inclusion. We also examined the association between polypharmacy and mortality using a longitudinal study design with the entire cohort included and followed from time of inclusion until end of follow-up. An overview of the study design and flow chart of the study population is illustrated in Fig 1. The correlations between different medications were examined among the polypharmacy-exposed population at time of inclusion.

### Assessment of polypharmacy

While there is no consensus definition of polypharmacy, it is frequently defined as the concurrent use of at least five medications. We defined polypharmacy as a monthly intake of at least five different medications, while excessive polypharmacy was defined as a monthly intake of at least 10 medications. Medications were defined by distinct substances according to the fifth level of the Anatomical Therapeutic Chemical (ATC) classification system. Data on polypharmacy were retrieved from the Danish National Prescription Registry. To calculate the monthly intake of medications, information on the prescribed amount of the medication as well as the defined daily dose (DDD) determined by WHO or national guidelines in Denmark were used [16]. As DDD is a central part of calculating the exposure to polypharmacy, those medications that did not have a DDD (4%) were excluded from the study. Prevalence was calculated based on information from the month prior to inclusion, while incidence was calculated from the time of inclusion until the end of follow-up.

### Covariates

Information on covariates were retrieved from the National Patient Registry, The Danish National Prescription Registry, The Population Register, The Education Register and The Income Register at the time of inclusion [17–21]. These covariates included sex, age, region of residence, country of birth, marital status, highest educational level, income, multimorbidity and year of inclusion.

We measured multimorbidity based on a list of 47 chronic conditions, which was selected according to recommendations in other studies [22–25]. Each of the chronic conditions was defined by ICD-10 codes, which should be present at least once in the National Patient Registry, and for some of the conditions, also ATC codes, which should be present at least twice in the Danish National Prescription Registry. In line with the framework developed by Hvidberg

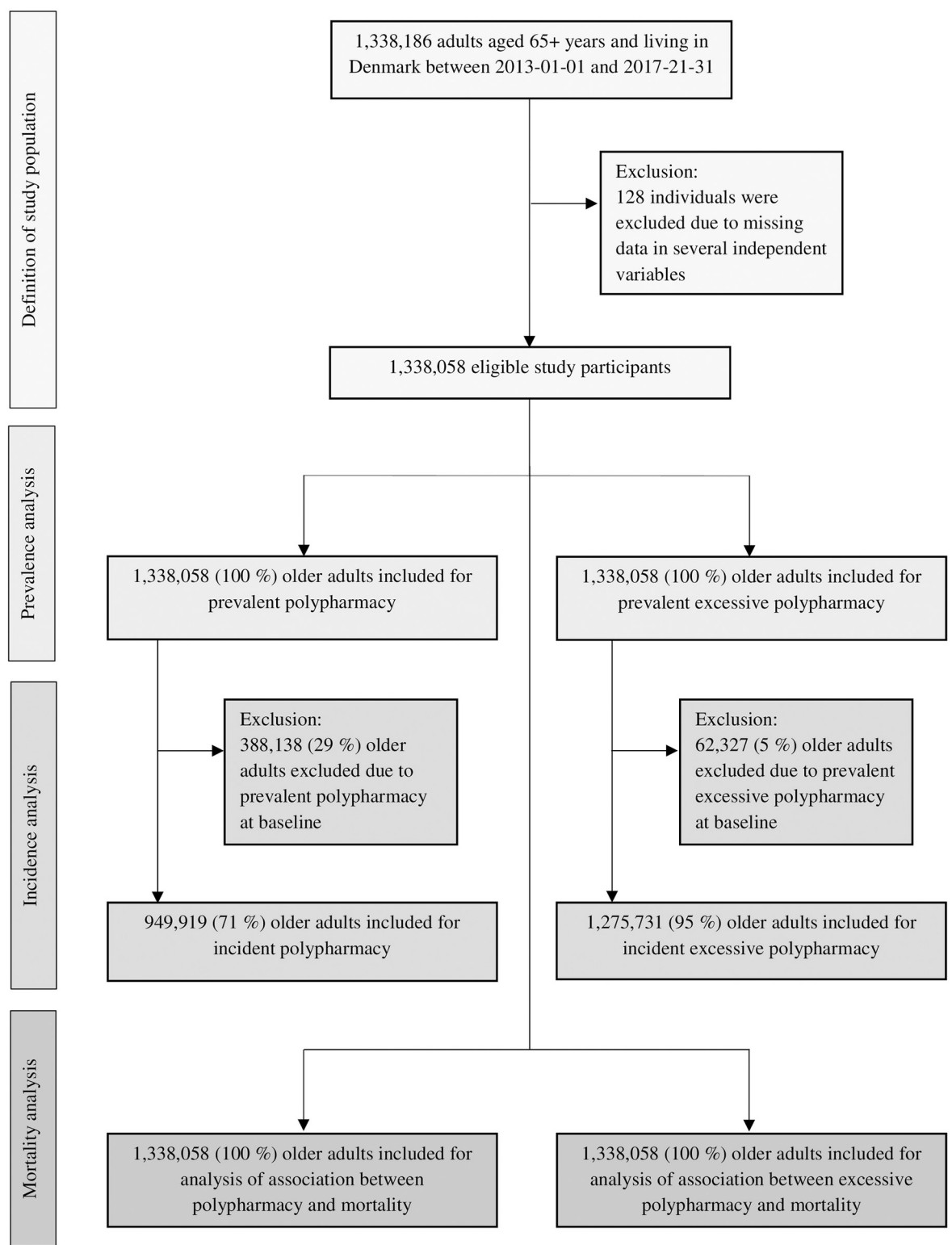

**Fig 1. Flow chart of study population and inclusion in each analysis.**

and colleagues [23], a time frame was specified for each condition. These time frames determined how many years before study entrance, information on ICD-10 codes should be drawn from the registers. The time frames fell into four categories depending on the expected duration of the chronic condition: 1) the entire life span (due to the introduction of ICD-10 in 1995, information were drawn from 1995 until study entrance), 2) 10 years before study entrance, 3) 5 years before study entrance and 4) 2 years before study entrance. Information on ATC codes were drawn two years prior to study entrance. The algorithm to define chronic conditions is presented in S1 Table.

## Statistical analysis

The prevalence of polypharmacy at time of inclusion were calculated as percentages of the total population. Adjusted logistic regression were used to identify factors associated with baseline exposure to polypharmacy and excessive polypharmacy.

Incidence rates of polypharmacy over the follow-up period were calculated as first time exposure to polypharmacy per 1,000 person-years. To identify factors associated with incident polypharmacy, we used Cox regression with time-on-study as the underlying time scale. Time at risk was counted from time of inclusion until the first occurrence of polypharmacy, death, migration or until the end of follow-up.

For the mortality analysis, polypharmacy status was treated as a time-dependent variable. Two different approaches for adjustment were used to identify the most appropriate method. The first approach was the conventional covariate adjustment, where covariates are simply added to the regression model. As the second approach, we used propensity score (PS) adjustment, where propensity scores were calculated for polypharmacy based on all covariates and included in the regression model. We used the overlap weighting (OW) method in the PS analyses [26]. The OW method assigns each participant a weight that represents their probability of being assigned to the opposite group [27]. As a result, participants who have a PS of 0.5, which indicates that the participants are equally likely to be assigned either polypharmacy or no polypharmacy, are up-weighted, while participants with extreme PS close to 0 and 1 are down-weighted. The main benefit of this approach is the way that it deals with extreme PS. In comparison, the commonly used inverse probability weighting (IPW) has limitations when dealing with extreme PS. Specifically, if there are cases where the value of covariates for which to probability of receiving the treatment is 0 or 1, the treated and control units at these values cannot be compared as this will result in biased estimates of the treatment effect [26]. Therefore, IPW is often combined with trimming in which participants in the tails of the PS distribution are excluded. The OW weights smoothly reduce the influence of participants with extreme PS and therefore trimming is unnecessary.

When using PS, the aim is to create balanced populations to increase comparability between the exposed and unexposed. Whether this was achieved was examined through descriptive analysis of the distribution of covariates among the polypharmacy-exposed and unexposed before and after PS weighting in S2 and S3 Tables. Overall, the population was perfectly balanced when introducing weights.

Among older adults who were exposed to polypharmacy at time of inclusion, we investigated which medications were most often taken in combination by estimating polychoric correlations and presenting these graphically in a heat map [28]. The correlations were calculated between therapeutic subgroups—the third level of the ATC classification system—to increase readability of the heat map. A list of therapeutic groups and their description is found in S4 Table.

We conducted a sensitivity analyses to examine the sensitivity of the definition for poly-pharmacy used in the present study. For this purpose, we excluded ATC codes for anti-infectives usually used for short-term treatments and products with no therapeutic purpose from the definition of polypharmacy. An overview of the excluded ATC codes are reported in S5 Table. Excluding ATC codes from the polypharmacy definition did not result in noteworthy changes to the mean medication use or to the prevalence and incidence estimates. Some descriptive results from the sensitivity analysis are presented in S6 Table.

The statistical software R version 3.6.1 was used for analysis [29–38].

### Ethics

According to Danish legislation no ethical approval is required for register-based studies. This study is registered with the Data Protection Agency via Statistics Denmark. Moreover, in Denmark formal agreements of informed consent from the participants are not needed for register-based studies.

## Results

In total 1,338,058 older adults aged 65+ years and living in Denmark were included in the study. Their characteristics at the time of inclusion are reported in Table 1.

### Prevalence

The average number of concurrent medications taken during the month prior to inclusion was 3.3 different medications. Mean medication use increased with age, so that those aged 90+ years had an average intake of five medications. In 2013 older adults in all age groups were included, while in 2014–2017 only adults who reached the age limit of 65 years entered the study population. This difference in age structure is therefore reflected in the marked difference in mean medication use in 2013 compared to the subsequent years. The mean 3.5 medication use were highest among people who were included in 2013, while it remained 2.4 across 2014–2017. An additional analysis conducted among only the 65 year-olds included in 2013 showed that the mean number of medications taken was 2.4, and 18.6% of the population were exposed to polypharmacy, while 2.5% were exposed to excessive polypharmacy (data not shown). Thus, medication use was similar among the same age group across the years studied.

At the time of inclusion, 29% had an intake of at least five medications, while almost 5% had an intake of at least 10 different medications (Table 2). The risk of polypharmacy were highest among people who were included in 2013 and the odds increased with higher age, leaving the 95+ year-olds with the highest OR of 4.54 (CI 95%: 4.31–4.79) compared to those aged 65–69 years.

As expected, the risk of polypharmacy was substantially higher among participants with multimorbidity, who had an OR of 26.6 (CI 95%: 26.2–27.1) compared to those with less than two chronic conditions. Those with a higher educational level or a higher income had lower levels of polypharmacy compared to those with lower educational levels or lower income. The same trends were found for excessive polypharmacy except for sex. Women had reduced odds for polypharmacy (OR = 0.96; CI 95%: 0.96–0.97), however, increased for excessive polypharmacy (OR = 1.13; CI 95%: 1.11–1.15).

### Incidence

Among older adults without polypharmacy at the time of inclusion, 445,947 (46.9%) transitioned into polypharmacy and 225,357 (17.7%) transitioned into excessive polypharmacy

**Table 1. Population characteristics at time of inclusion.**

| | Population characteristics | |
|---|---|---|
| | **N** | **%** |
| All | 1,338,058 | 100 |
| **Sex** | | |
| Male | 618,652 | 46.2 |
| Female | 719,406 | 53.8 |
| **Age** | | |
| 65–69 years | 687,797 | 51.4 |
| 70–74 years | 243,346 | 18.2 |
| 75–79 years | 173,789 | 13.0 |
| 80–84 years | 118,459 | 8.9 |
| 85–89 years | 73,998 | 5.5 |
| 90–94 years | 32,104 | 2.4 |
| 95+ years | 8,565 | 0.6 |
| **Region of residence** | | |
| Northern Jutland Region | 149,063 | 11.1 |
| Mid Jutland Region | 291,894 | 21.8 |
| Region of Southern Denmark | 304,176 | 22.7 |
| Capital Region of Denmark | 373,042 | 27.9 |
| Region Zealand | 219,883 | 16.4 |
| **Region of birth** | | |
| Denmark | 1,273,965 | 95.2 |
| Other western country | 36,615 | 2.7 |
| Non-Western country | 27,577 | 2.1 |
| **Marital Status** | | |
| Married | 789,565 | 59.0 |
| Divorced | 179,374 | 13.4 |
| Widowed | 282,021 | 21.1 |
| Never married | 87,098 | 6.5 |
| **Highest achieved education** | | |
| No education | 46,634 | 3.5 |
| Secondary school | 501,155 | 37.5 |
| High school/skilled education | 510,095 | 38.1 |
| Short higher education | 37,774 | 2.8 |
| Middle higher education | 177,027 | 13.2 |
| High higher education | 65,373 | 4.9 |
| **Income** | | |
| First quartile (lowest) | 333,552 | 24.9 |
| Second quartile | 333,544 | 24.9 |
| Third quartile | 333,545 | 24.9 |
| Fourth quartile (highest) | 333,547 | 24.9 |
| Unknown | 3,870 | 0.4 |
| **Number of chronic conditions** | | |
| 0–1 | 522,269 | 39.0 |
| 2+ | 815,789 | 61.0 |
| **Year of inclusion** | | |
| 2013 | 1,070,562 | 80.0 |
| 2014 | 67,582 | 5.1 |

(*Continued*)

**Table 1.** (Continued)

| | Population characteristics | |
|---|---|---|
| | **N** | **%** |
| 2015 | 67,776 | 5.1 |
| 2016 | 65,596 | 4.9 |
| 2017 | 66,542 | 5.0 |

during follow-up (Tables 3 & 4). The rate of transitioning into polypharmacy increased markedly with later year of inclusion. Older adults who were included in the study in 2017 had an incidence rate of 261 cases per 1,000 person-years, whereas it was 134 cases in 2014. The hazard increased with age, leaving the 95+ year-olds with an increased hazard of 3.10 (CI 95%: 2.74–2.96) compared to the 65–69 year-olds.

The rate of transitioning into polypharmacy was markedly higher among older adults with multimorbidity. Hence, older adults with 2+ chronic conditions had an increased hazard for transitioning into polypharmacy of 3.51 (CI 95%: 3.48–3.53) compared to older adults who did not have any chronic conditions. The results also indicate an association between marital status and polypharmacy in which older adults who were either divorced (HR = 1.11; CI 95%: 1.10–1.12) or widowed (HR = 1.11; CI 95%: 1.11–1.12) had an increased hazard compared to those who were married. Regarding socioeconomic factors, the hazard for transitioning into polypharmacy decreased with higher completed education and with higher income. Similar trends were observed for excessive polypharmacy. The association between region of birth did, however, differ. Being a migrant from non-Western countries were associated with a reduced hazard for transitioning into polypharmacy (HR = 0.77; CI 95%: 0.75–0.79), while it was associated with an increased hazard for transitioning into excessive polypharmacy (HR = 1.07; CI 95%: 1.03–1.10).

## Mortality

As reported in Table 5, there was a strong association between polypharmacy and mortality. When adjusting for covariates, older adults exposed to polypharmacy had an increased hazard of 3.95 (CI 95%: 3.90–4.01) for death compared to those who were not exposed to polypharmacy, while they had an increased hazard of 3.48 (CI 95%: 3.41–3.54) when adjusting for weighted PS. For excessive polypharmacy, a similar trend was observed.

## Drug cocktails

Older adults exposed to polypharmacy at baseline most often took medication for the cardiovascular system (95%), blood and blood-forming organs (69%), alimentary tract and metabolism (61%) and nervous system (54%) (S4 Table). Fig 2 presents correlations between therapeutic subgroups. Among the negative correlations were the therapeutic subgroups C07-C10 (cardiovascular medication), which indicates that patients are typically treated with medications from one subgroup, not a cocktail across subgroups. Many of the positive correlations reflect that different medications are used to treat the same underlying condition, e.g. the correlation between C03 (diuretics), which is given to treat certain kidney disorders and A12 (mineral supplements), which is given to for disturbed homeostasis of minerals.

## Discussion

A considerable high level of polypharmacy was found among older adults in Denmark: i.e. almost one third of older adults were exposed to polypharmacy at time of inclusion, while

**Table 2. Prevalence and risk of being exposed to polypharmacy and excessive polypharmacy at the time of inclusion (N = 1,338,058).**

| | Mean medication use | Polypharmacy | | | | Excessive polypharmacy | | | |
|---|---|---|---|---|---|---|---|---|---|
| | | Prevalence | | Model I[a] | | Prevalence | | Model I[a] | |
| | | N | % | OR | CI 95% | N | % | OR | CI 95% |
| **Overall** | 3.3 | 388,138 | 29.0 | | | 62,327 | 4.7 | | |
| **Sex** | | | | | | | | | |
| Male | 3.1 | 170,539 | 27.7 | Ref. | | 24,954 | 4.0 | Ref. | |
| Female | 3.4 | 217,599 | 30.2 | 0.96 | 0.96–0.97 | 37,373 | 5.2 | 1.13 | 1.11–1.15 |
| **Age** | | | | | | | | | |
| 65–69 years | 2.5 | 139,085 | 20.2 | Ref. | | 19,455 | 2.8 | Ref. | |
| 70–74 years | 3.3 | 70,529 | 29.0 | 1.40 | 1.38–1.42 | 10,836 | 4.5 | 1.35 | 1.32–1.39 |
| 75–79 years | 4.0 | 65,790 | 37.9 | 1.90 | 1.88–1.92 | 11,193 | 6.4 | 1.78 | 1.73–1.82 |
| 80–84 years | 4.5 | 53,647 | 45.3 | 2.47 | 2.43–2.50 | 9,880 | 8.3 | 2.19 | 2.14–2.25 |
| 85–89 years | 4.9 | 37,554 | 50.8 | 3.11 | 3.05–3.16 | 7,113 | 9.6 | 2.48 | 2.41–2.56 |
| 90–94 years | 5.1 | 17,015 | 53.0 | 3.71 | 3.61–3.81 | 3,143 | 9.8 | 2.58 | 2.47–2.68 |
| 95+ years | 5.0 | 4,518 | 52.7 | 4.54 | 4.31–4.79 | 707 | 8.3 | 2.30 | 1.13–2.49 |
| **Region of residence** | | | | | | | | | |
| Northern Jutland Region | 3.5 | 47,496 | 31.9 | Ref. | | 8,272 | 5.5 | Ref. | |
| Mid Jutland Region | 3.4 | 88,188 | 30.2 | 0.92 | 0.91–0.94 | 15,314 | 5.2 | 0.96 | 0.93–0.98 |
| Region of Southern Denmark | 3.3 | 90,957 | 29.9 | 0.87 | 0.85–0.88 | 14,626 | 4.8 | 0.84 | 0.82–0.87 |
| Capital Region of Denmark | 3.1 | 100,581 | 27.0 | 0.77 | 0.75–0.78 | 14,781 | 4.0 | 0.71 | 0.69–0.73 |
| Region Zealand | 3.2 | 60,916 | 27.7 | 0.87 | 0.85–0.88 | 9,334 | 4.2 | 0.81 | 0.79–0.84 |
| **Migration status** | | | | | | | | | |
| Danish | 3.3 | 374,067 | 29.4 | Ref. | | 60,044 | 4.7 | Ref. | |
| Western migrant | 2.4 | 7,563 | 20.7 | 0.73 | 0.73–0.70 | 1,210 | 3.3 | 0.82 | 0.78–0.87 |
| Non-Western migrant | 2.6 | 6,508 | 23.6 | 0.89 | 0.89–0.86 | 1,073 | 3.9 | 1.03 | 0.97–1.10 |
| **Marital Status** | | | | | | | | | |
| Married | 2.9 | 194,920 | 24.7 | Ref. | | 26,883 | 3.4 | Ref. | |
| Divorced | 3.4 | 54,400 | 30.3 | 1.32 | 1.30–1.35 | 10,345 | 7.6 | 1.65 | 1.61–1.69 |
| Widowed | 4.2 | 115,878 | 41.1 | 1.33 | 1.31–1.35 | 21,359 | 5.8 | 1.43 | 1.40–1.46 |
| Never married | 2.9 | 22,940 | 26.3 | 1.24 | 1.22–1.26 | 3,740 | 4.3 | 1.38 | 1.33–1.43 |
| **Highest achieved education** | | | | | | | | | |
| No education | 3.6 | 16,544 | 35.5 | Ref. | | 2,816 | 6.0 | Ref. | |
| Secondary school | 3.9 | 182,451 | 36.4 | 1.44 | 1.40–1.48 | 33,581 | 6.7 | 1.29 | 1.23–1.36 |
| High school/skilled education | 3.0 | 133,873 | 26.2 | 1.09 | 1.06–1.13 | 19,067 | 3.7 | 0.89 | 0.85–0.94 |
| Short higher education | 2.6 | 8,103 | 21.5 | 0.92 | 0.88–0.96 | 990 | 2.6 | 0.69 | 0.64–0.75 |
| Middle higher education | 2.6 | 35,894 | 20.3 | 0.85 | 0.82–0.87 | 4,551 | 2.6 | 0.67 | 0.63–0.71 |
| High higher education | 2.3 | 11,273 | 17.2 | 0.75 | 0.72–0.77 | 1,322 | 2.0 | 0.59 | 0.55–0.63 |
| **Income** | | | | | | | | | |
| First quartile (lowest) | 4.0 | 128,578 | 38.5 | Ref. | | 23,692 | 7.1 | Ref. | |
| Second quartile | 3.8 | 117,875 | 35.3 | 0.94 | 0.93–0.96 | 21,086 | 6.3 | 0.96 | 0.94–0.98 |
| Third quartile | 3.0 | 84,381 | 25.3 | 0.70 | 0.69–0.71 | 11,646 | 3.5 | 0.63 | 0.61–0.64 |
| Fourth quartile (highest) | 2.3 | 56,715 | 17.0 | 0.50 | 0.50–0.51 | 5,782 | 1.7 | 0.38 | 0.37–0.39 |
| Unknown | 1.8 | 589 | 15.2 | 0.69 | 0.62–0.76 | 121 | 3.1 | 1.06 | 0.87–1.27 |
| **Number of chronic conditions** | | | | | | | | | |
| 0–1 | 1.1 | 14,775 | 2.8 | Ref. | | 323 | 0,1 | Ref. | |
| 2+ | 4.7 | 373,363 | 45.8 | 26.64 | 26.20–27.10 | 62,004 | 7.6 | 116.07 | 104.24–129.76 |
| **Year of inclusion[b]** | | | | | | | | | |
| 2013 | 3.5 | 338,939 | 31.7 | Ref. | | 55,505 | 5.2 | Ref. | |
| 2014 | 2.4 | 12,259 | 18.1 | 0.86 | 0.84–0.88 | 1,724 | 2.6 | 0.91 | 0.86–0.96 |

*(Continued)*

**Table 2.** (Continued)

| | Mean medication use | Polypharmacy | | | | Excessive polypharmacy | | | |
|---|---|---|---|---|---|---|---|---|---|
| | | Prevalence | | Model I[a] | | Prevalence | | Model I[a] | |
| | | N | % | OR | CI 95% | N | % | OR | CI 95% |
| 2015 | 2.4 | 12,552 | 18.5 | 0.88 | 0.86–0.90 | 1,714 | 2.5 | 0.89 | 0.85–0.94 |
| 2016 | 2.4 | 12,214 | 18.6 | 0.87 | 0.85–0.89 | 1,707 | 2.6 | 0.91 | 0.86–0.96 |
| 2017 | 2.4 | 12,174 | 18.3 | 0.79 | 0.76–0.81 | 1,677 | 2.5 | 0.93 | 0.79–0.88 |

[a] Adjusted for sex, age, and number of chronic conditions

[b] Note: The age structure is different in 2013 compared to the subsequent years due to the inclusion criteria. In 2013 older adults in all age groups are included, while in 2014–2017 only adults who reach the age limit of 65 years enter the study population. This difference in age structure is reflected in the marked difference in mean medication use in 2013 compared to 2014–2017.

more than five percent were exposed to excessive polypharmacy. Over a five-year period, 47 percent transitioned into polypharmacy, while 18 percent transitioned into excessive polypharmacy. Age and year of inclusion were identified as important predictors for polypharmacy, as the risk of transitioning into polypharmacy increased markedly with age and with later year of inclusion. Moreover, multimorbidity, socioeconomic factors, region of birth, region of residence and marital status were identified risk factors. Both polypharmacy and excessive polypharmacy increased the risk of premature death three- to fourfold.

## Polypharmacy occurrence

The high prevalence of polypharmacy (29%) found in this study confirms findings from other studies, however, the estimate is among the lowest prevalence estimates [5, 8, 9, 39, 40]. Moriarty et al. [8] found that 60% of adults aged 65+ years in Ireland were exposed to polypharmacy in 2012. Similarly, Franchi et al. [39] found that 53% of adults aged 65–94 years living in the Lombardy region of Italy were exposed to polypharmacy in 2010. However, Moriarty et al. [8] and Franchi [39] defined polypharmacy as the use of at least five medications within a year whereas the present study defines polypharmacy based on monthly medication use. Hovstadius and colleagues [9] defined polypharmacy as the concurrent use of five medications over three months and found that polypharmacy was prevalent among 41% of the 60+ years-olds living in Sweden in 2008. The differences in estimates can be caused by various contextual factors; however, the differences in how polypharmacy is defined is expected to be the primary cause. A more recent study conducted in Sweden by Morin and colleagues [5] used a definition and study design more comparable to the design of the present study and found a prevalence of 44% among 65+ year-old in Sweden.

To our knowledge, only few studies have examined the incidence of polypharmacy [5, 41–43]. The present study, therefore, contributes to filling a gap in the knowledge of polypharmacy occurrence over time. The incidence rates found in this study shows that 172 older adults transitioned into polypharmacy in 1,000 person-years. In comparison, Morin and colleagues [5] found a similar incidence rate of 199 cases in 1,000 person-years. The differences in the incidence rates may be due to differences in available information as the prescribed intake duration is registered in the Swedish registers, while it is not in Denmark.

## What drives the high occurrence of polypharmacy?

The high incidence of polypharmacy among older adults is fuelled by multiple factors including the increasing demand for drug therapy in advanced ages and the development of new

**Table 3. Incidence of polypharmacy among the 949,919 study participants who were not exposed to polypharmacy at the time of inclusion.**

| | Person-years | Incident cases | Incidence rate | Model I[a] | | Model II[b] | |
|---|---|---|---|---|---|---|---|
| | N | N | Pr. 1,000 person-years | HR | CI 95% | HR | CI 95% |
| **Overall** | 2,597,466 | 445,947 | 172 | | | | |
| **Sex** | | | | | | | |
| Male | 1,235,857 | 205,585 | 166 | Ref. | | Ref. | |
| Female | 1,361,609 | 240,362 | 177 | 1.06 | 1.05–1.07 | 0.97 | 0.97–0.98 |
| **Age** | | | | | | | |
| 65–69 years | 1,499,755 | 202,457 | 135 | Ref. | | Ref. | |
| 70–74 years | 542,742 | 93,018 | 171 | 1.31 | 1.30–1.32 | 1.22 | 1.21–1.23 |
| 75–79 years | 297,907 | 66,838 | 224 | 1.68 | 1.66–1.69 | 1.48 | 1.47–1.49 |
| 80–84 years | 153,395 | 44,338 | 289 | 2.10 | 2.08–2.12 | 1.80 | 1.78–1.82 |
| 85–89 years | 73,149 | 26,065 | 356 | 2.50 | 2.47–2.53 | 2.13 | 2.11–2.16 |
| 90–94 years | 25,213 | 10,639 | 422 | 2.84 | 2.79–2.90 | 2.49 | 2.44–2.54 |
| 95+ years | 5,306 | 2,592 | 489 | 3.10 | 2.98–3.22 | 2.85 | 2.74–2.96 |
| **Region of residence** | | | | | | | |
| Northern Jutland Region | 275,936 | 48,129 | 174 | Ref. | | Ref. | |
| Mid Jutland Region | 555,326 | 95,403 | 172 | 0.98 | 0.97–1.00 | 0.98 | 0.97–0.99 |
| Region of Southern Denmark | 584,401 | 100,321 | 172 | 0.99 | 0.98–1.00 | 0.95 | 0.94–0.96 |
| Capital region of Denmark | 743,940 | 127,663 | 171 | 0.99 | 0.97–1.00 | 0.96 | 0.95–0.97 |
| Region Zealand | 437,863 | 74,431 | 170 | 0.98 | 0.97–0.99 | 1.00 | 0.99–1.01 |
| **Region of birth** | | | | | | | |
| Danish | 2,459,241 | 428,738 | 174 | Ref. | | Ref. | |
| Western migrant | 82,674 | 9,919 | 120 | 0.69 | 0.68–0.71 | 0.73 | 0.72–0.75 |
| Non-Western migrant | 55,551 | 7,290 | 13 | 0.75 | 0.73–0.77 | 0.77 | 0.75–0.79 |
| **Marital Status** | | | | | | | |
| Married | 1,682,127 | 264,817 | 157 | Ref. | | Ref. | |
| Divorced | 417,295 | 98,667 | 236 | 1.11 | 1.10–1.12 | 1.11 | 1.10–1.12 |
| Widowed | 327,878 | 58,342 | 178 | 1.48 | 1.47–1.49 | 1.11 | 1.11–1.12 |
| Never married | 170,066 | 24,121 | 142 | 0.89 | 0.88–0.90 | 0.94 | 0.93–0.95 |
| **Highest achieved education** | | | | | | | |
| No education | 76,023 | 13,236 | 174 | Ref. | | Ref. | |
| Secondary school | 848,800 | 172,838 | 203 | 1.18 | 1.16–1.20 | 1.48 | 1.48–1.55 |
| High school/skilled education | 1,033,432 | 172,838 | 167 | 0.96 | 0.94–0.98 | 1.36 | 1.36–1.42 |
| Short higher education | 84,065 | 12,043 | 143 | 0.83 | 0.81–0.86 | 1.23 | 1.23–1.30 |
| Middle higher education | 399,908 | 56,784 | 142 | 0.83 | 0.81–0.84 | 1.21 | 1.21–1.27 |
| High higher education | 155,238 | 20,255 | 130 | 0.76 | 0.75–0.78 | 1.17 | 1.17–1.22 |
| **Income** | | | | | | | |
| First quartile | 538,772 | 114,719 | 213 | Ref. | | Ref. | |
| Second quartile | 584,240 | 115,665 | 198 | 0.93 | 0.93–0.94 | 0.99 | 0.98–1.00 |
| Third quartile | 701,146 | 112,074 | 160 | 0.76 | 0.75–0.76 | 0.88 | 0.88–0.89 |
| Fourth quartile | 766,962 | 102,224 | 133 | 0.63 | 0.62–0.63 | 0.81 | 0.80–0.81 |
| Unknown | 6,347 | 1,265 | 199 | 0.88 | 0.83–0.93 | 1.33 | 1.26–1.40 |
| **Number of chronic conditions** | | | | | | | |
| 0–1 | 1,710,105 | 146,533 | 86 | Ref. | | Ref. | |
| 2+ | 887,361 | 299,414 | 337 | 3.67 | 3.65–3.69 | 3.51 | 3.48–3.53 |
| **Year of inclusion[c]** | | | | | | | |
| 2013 | 2,245,482 | 391,061 | 174 | Ref. | | Ref. | |
| 2014 | 147,329 | 19,695 | 134 | 0.72 | 0.71–0.73 | 0.99 | 0.98–1.01 |

*(Continued)*

**Table 3.** (Continued)

| | Person-years | Incident cases | Incidence rate | Model I[a] | | Model II[b] | |
|---|---|---|---|---|---|---|---|
| | N | N | Pr. 1,000 person-years | HR | CI 95% | HR | CI 95% |
| 2015 | 110,654 | 16,694 | 151 | 0.73 | 0.72–0.74 | 1.02 | 1.00–1.04 |
| 2016 | 68,925 | 11,951 | 173 | 0.72 | 0.70–0.73 | 1.00 | 0.98–1.02 |
| 2017 | 25,076 | 6,546 | 261 | 0.86 | 0.84–0.89 | 1.19 | 1.19–1.22 |

[a] Unadjusted

[b] Adjusted for sex, age and number of chronic conditions

[c] Note: The age structure is different in 2013 compared to the subsequent years due to the inclusion criteria. In 2013, older adults in all age groups were included, while in 2014–2017 only adults who reached the age limit of 65 years entered the study population.

medications. We investigated the incidence in an open cohort in which the estimates both reflect an age effect and a period effect. The period effect is entangled with the cohort effect, however, we argue that the cohort effect does not play a significant role in the polypharmacy incidence calculated in the present study. Health has improved in all age groups and in particular among older adults over time leading to a higher life expectancy [1]. Consequently, the need for medication should decrease with new cohorts and therefore not fuel the high incidence of polypharmacy. We thus assume that the occurrence of polypharmacy are primarily driven by age and period effects. Age effects entail that older adults are growing older during the follow-up time, during which time they may develop chronic conditions, which will require drug therapy. Conversely, we found that the risk of transitioning into polypharmacy increased with higher age. The risk of polypharmacy increased through all age groups, which contradicts findings from other studies as the increase tends to stagnate after the age of 85 years [44, 45]. However, a study by Wastesson et al. [46] found that medicine use continues to increase in the oldest old, just at a much lower rate. This finding partly corresponds to the findings for excessive polypharmacy as incidence rates begin to stagnate from age 85–89 years and even decrease in the oldest age groups. Furthermore, we investigated the period effect, which reflects that the risk of being exposed to polypharmacy changes over time possibly due to changes in prescription guidelines, the development of new medication and expansion of treatment in older age. Several studies have investigated the prescription of medications over time and found that the prescription of specific medications increased a long with the overall medication intake [41, 47–49]. This could be an indication that prescription patterns are changing and more medications are being prescribed. Conversely, we found that the polypharmacy incidence rate increased markedly from 2014 to 2017. This finding supports the notion that being exposed to polypharmacy is not only driven by the health or illness of the individual but also by societal factors such as treatment guidelines and the development of new medication; thus indicating the importance of period effects. We found the highest mean medication intake among older adults who were included in 2013, however, this is most likely because the majority of the study participants—including the oldest population—were included in this year, while for the later years only people who turned 65 years or who migrated to Denmark were included. Therefore, the period effect are best observed from 2014 to 2017. To limit the effect of different age structures across the years, we adjusted for age in the Cox regression model II. The results show that when adjusting for age as well as sex and multimorbidity the hazard of transitioning into polypharmacy is not significant across the years of inclusion with the only exception of 2017.

Furthermore, we identified risk factors for polypharmacy. We found, older adults with multimorbidity was a high-risk group for being exposed to polypharmacy. This is not surprising as

**Table 4. Incidence of excessive polypharmacy among the 1,275,731 study participants were not exposed to excessive polypharmacy at the time of inclusion.**

| | Person-years | Incident cases | Incidence rate | Model I[a] | | Model II[b] | |
|---|---|---|---|---|---|---|---|
| | N | N | Pr. 1,000 person-years | HR | CI 95% | HR | CI 95% |
| **Overall** | 4,605,409 | 225,357 | 49 | | | | |
| **Sex** | | | | | | | |
| Male | 2,134,595 | 99,197 | 46 | Ref. | | Ref. | |
| Female | 2,470,814 | 126,160 | 5 | 1.10 | 1.09–1.10 | 0.97 | 0.96–0.97 |
| **Age** | | | | | | | |
| 65–69 years | 2,307,335 | 70,302 | 30 | Ref. | | Ref | |
| 70–74 years | 989,209 | 44,532 | 45 | 1.52 | 1.50–1.54 | 1.34 | 1.32–1.36 |
| 75–79 years | 638,615 | 42,127 | 66 | 2.21 | 2.18–2.23 | 1.79 | 1.77–1.81 |
| 80–84 years | 380,202 | 34,514 | 91 | 3.00 | 2.96–3.04 | 2.32 | 2.29–2.35 |
| 85–89 years | 202,514 | 22,784 | 113 | 3.64 | 3.59–3.70 | 2.80 | 2.75–2.84 |
| 90–94 years | 72,238 | 9,188 | 127 | 4.00 | 3.92–4.09 | 3.18 | 3.11–3.25 |
| 95+ years | 15,296 | 1,910 | 12 | 3.79 | 3.62–3.97 | 3.21 | 3.07–3.36 |
| **Region of residence** | | | | | | | |
| Northern Jutland Region | 505,760 | 26,126 | 517 | Ref. | | Ref. | |
| Mid Jutland Region | 989,671 | 51,822 | 52 | 1.01 | 1.00–1.03 | 1.03 | 1.01–1.04 |
| Region of Southern Denmark | 1,049,117 | 51,238 | 49 | 0.95 | 0.93–0.96 | 0.93 | 0.92–0.94 |
| Capital region of Denmark | 1,296,102 | 60,929 | 47 | 0.91 | 0.90–0.92 | 0.92 | 0.91–0.93 |
| Region Zealand | 764,760 | 35,242 | 461 | 0.89 | 0.88–0.91 | 0.95 | 0.94–0.97 |
| **Region of birth** | | | | | | | |
| Danish | 4,393,137 | 216,963 | 49 | Ref. | | Ref. | |
| Western migrant | 125,345 | 4,501 | 36 | 0.72 | 0.70–0.75 | 0.82 | 0.79–0.84 |
| Non-Western migrant | 86,928 | 3,893 | 45 | 0.89 | 0.87–0.92 | 1.07 | 1.03–1.10 |
| **Marital Status** | | | | | | | |
| Married | 2,843,656 | 112,275 | 39 | Ref. | | Ref. | |
| Divorced | 582,436 | 32,023 | 55 | 1.38 | 1.36–1.40 | 1.37 | 1.35–1.39 |
| Widowed | 906,746 | 69,012 | 76 | 1.92 | 1.90–1.93 | 1.25 | 1.24–1.27 |
| Never married | 272,571 | 12,047 | 44 | 1.10 | 1.08–1.12 | 1.17 | 1.15–1.19 |
| **Highest achieved education** | | | | | | | |
| No education | 131,869 | 8,699 | 66 | Ref. | | Ref. | |
| Secondary school | 1,688,828 | 118,221 | 70 | 0.99 | 0.97–1.02 | 1.24 | 1.21–1.27 |
| High school/skilled education | 1,784,202 | 76,601 | 43 | 0.66 | 0.65–0.68 | 1.01 | 0.98–1.04 |
| Short higher education | 134,187 | 4,643 | 35 | 0.54 | 0.52–0.56 | 0.88 | 0.85–0.92 |
| Middle higher education | 632,575 | 20,504 | 32 | 0.50 | 0.49–0.52 | 0.83 | 0.80–0.85 |
| High higher education | 233,748 | 6,689 | 29 | 0.44 | 0.43–0.46 | 0.77 | 0.75–0.80 |
| **Income** | | | | | | | |
| First quartile | 1,109,136 | 78,741 | 71 | Ref. | | Ref. | |
| Second quartile | 1,139,606 | 68,095 | 60 | 0.84 | 0.83–0.85 | 0.93 | 0.92–0.94 |
| Third quartile | 1,186,754 | 47,404 | 40 | 0.56 | 0.56–0.57 | 0.73 | 0.72–0.74 |
| Fourth quartile | 1,160,442 | 30,710 | 3 | 0.37 | 0.37–0.38 | 0.57 | 0.56–0.58 |
| Unknown | 9,472 | 407 | 43 | 0.58 | 0.53–0.64 | 1.17 | 1.06–1.29 |
| **Number of chronic conditions** | | | | | | | |
| 0–1 | 2,069,898 | 21,195 | 10 | Ref. | | Ref. | |
| 2+ | 2,535,512 | 204,162 | 81 | 7.75 | 7.64–7.86 | 6.92 | 6.83–7.02 |
| **Year of inclusion[c]** | | | | | | | |
| 2013 | 4,108,542 | 209,284 | 51 | Ref. | | Ref. | |
| 2014 | 214,838 | 6,145 | 29 | 0.53 | 0.51–0.54 | 0.95 | 0.92–0.97 |

(*Continued*)

**Table 4.** (Continued)

|  | Person-years | Incident cases | Incidence rate | Model I[a] | | Model II[b] | |
|---|---|---|---|---|---|---|---|
|  | N | N | Pr. 1,000 person-years | HR | CI 95% | HR | CI 95% |
| 2015 | 156,832 | 4,891 | 31 | 0.53 | 0.51–0.54 | 0.96 | 0.93–0.99 |
| 2016 | 92,999 | 3,381 | 36 | 0.54 | 0.52–0.56 | 0.99 | 0.95–1.02 |
| 2017 | 32,199 | 1,656 | 51 | 0.68 | 0.65–0.71 | 1.21 | 1.15–1.27 |

[a] Unadjusted

[b] Adjusted for sex, age and number of chronic conditions

[c] Note: The age structure is different in 2013 compared to the subsequent years due to the inclusion criteria. In 2013 older adults in all age groups were included, while in 2014–2017 only adults who reached the age limit of 65 years entered the study population.

morbidity is the primary indicator for initiating drug therapy. Furthermore, high completed educational levels and income were associated with low risk of polypharmacy incidence. A substantial amount of studies have connected socioeconomic factors to health, thus as health is an indicator for polypharmacy, the results from the present study was expected [50]. Being married were associated with a reduced risk of polypharmacy compared to being either divorced or widowed, which may indicate that social relations and connectedness influence health behaviours and health care utilisation influencing medication intake [51].

In regards to region of birth, older adults born in Western countries had a reduced risk of developing either polypharmacy or excessive polypharmacy, while older adults born in non-Western countries had a reduced risk of polypharmacy and an increased risk of excessive poly-pharmacy compared to Danish-born. That migrants from non-Western countries have an increased risk of excessive polypharmacy may conflict with existing knowledge of low health care utilisation, language and cultural barriers as well as disadvantaged socioeconomic circum-stances that may restrict migrants from having medications prescribed [52]. However, non-Western migrants are found to have a higher risk of certain chronic conditions than native-born, which most often require drug therapy with multiple medications [53, 54]. Hence, our results may indicate that non-Western migrants in Denmark overall are less at risk for poly-pharmacy, however, a subgroup of migrants has a high medication use.

## Mortality

We found that both polypharmacy and excessive polypharmacy are associated with a substan-tial increase in the risk of premature death among older adults in Denmark. These associations

**Table 5. Association between polypharmacy status and mortality among the entire population (N = 1,338,058).**

|  | Model I[a] | | Model II[b] | | Model III[c] | | Model III[d] | | Model III[e] | |
|---|---|---|---|---|---|---|---|---|---|---|
|  | HR | CI 95% | HR | CI 95% | HR | CI 95% | HR | CI 95% | HR | CI 95% |
| No polypharmacy | Ref. | | Ref. | | Ref. | | Ref. | | Ref. | |
| Polypharmacy | 6.42 | 6.34–6.50 | 4.26 | 4.21–4.31 | 3.95 | 3.90–4.01 | 3.28 | 3.24–3.33 | 3.48 | 3.41–3.54 |
| No excessive polypharmacy | Ref. | | Ref. | | Ref. | | Ref. | | Ref. | |
| Excessive polypharmacy | 6.11 | 6.06–6.16 | 3.70 | 3.66–3.73 | 3.60 | 3.57–3.63 | 3.53 | 3.50–3.56 | 3.48 | 3.43–3.53 |

[a] Unadjusted

[b] Adjusted for sex, age, and number of chronic conditions

[c] Adjusted for sex, age, region of residence, region of birth, marital status, education, income, number of chronic conditions and year of inclusion.

[d] Adjusted for propensity score based on all covariates

[e] Adjusted for propensity score based on all covariates and weighted according to the overlap weighting method.

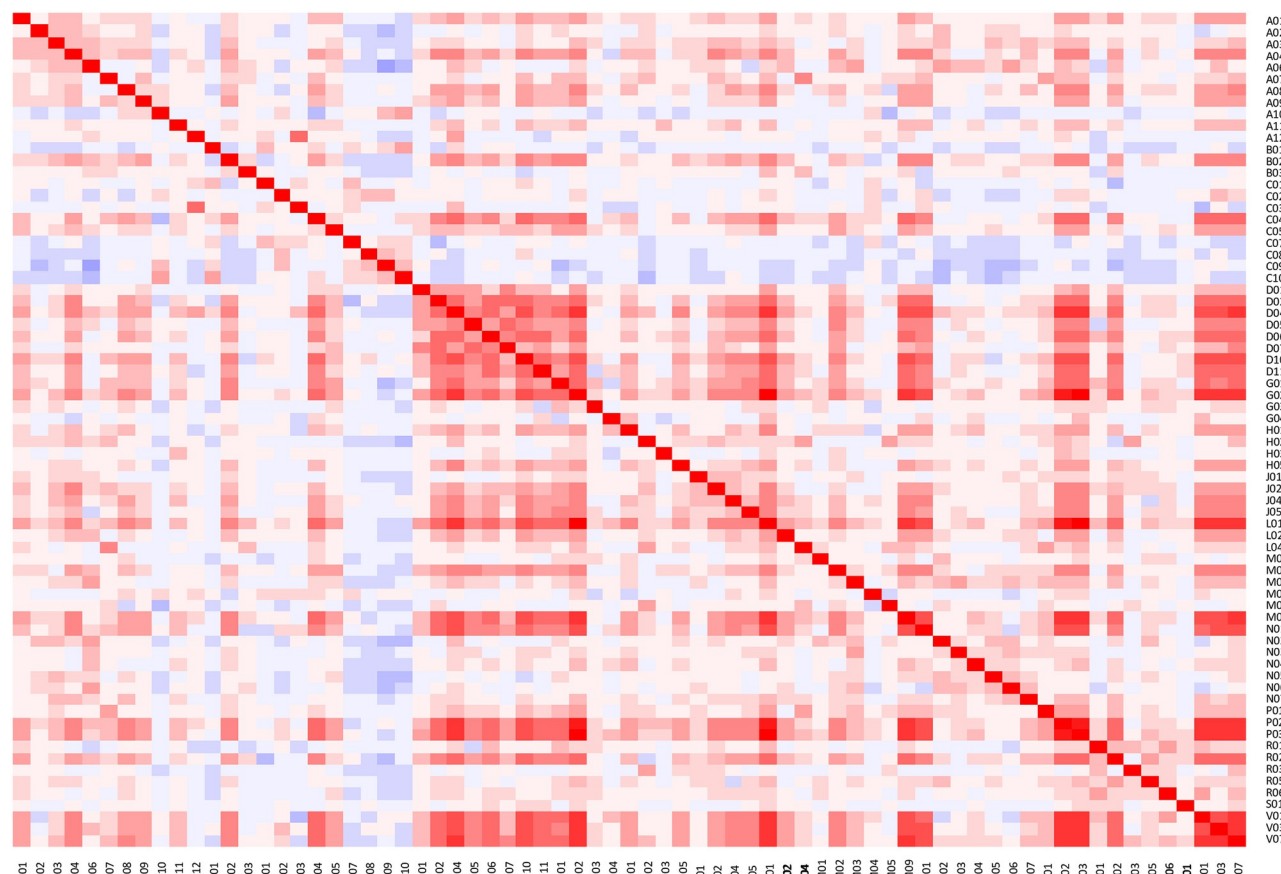

**Fig 2. Heatmap of correlations between medications among polypharmacy-exposed participants at the time of inclusion.**

may be explained by confounding by indication or mechanisms like drug-drug interactions, drug-disease interactions, and non-adherence to drug therapy [10, 11, 55]. It has been widely discussed how to properly adjust for confounding by indication when examining the association between polypharmacy and mortality. Consequently, we used two different approaches for adjustment when examining the association: the conventional approach for adjustment of potential confounders and adjustment for weighted PS. The association between polypharmacy and mortality was clear using both approaches; however, when adjusting for PS the hazard ratios were reduced to a higher extent than when adjusting for covariates. This may reflect that adjustment using PS was more successful than adjusting for confounding, as we expect that higher attenuation indicate better control for the substantial confounding from unobserved factors that is likely present. Nonetheless, it should be noted that the appropriateness of using PS in observational studies has been widely discussed following the increase in its popularity. The main aim of this method is to emulate randomised assignment to the treatment and control group [26]. As a result, the characteristics of the exposed and unexposed should be the same ideally resulting in an unbiased estimate of the treatment effect. Though, this relies on the assumption that there are no unmeasured confounders. This may not be the case in the present study, however, residual confounding may in that case affect both the estimates adjusted for PS and those adjusted for covariates. The strength of using PS accompanied with the overlap weighting method is, the perfectly balanced distribution of covariates across

exposure groups (S2 & S3 Tables). This strength makes the estimate adjusted for PS with weights the most reliable estimate to describe the association between polypharmacy and mortality.

To our knowledge, only one other study by Schöttker et al. [56] has examined the association between polypharmacy and mortality while adjusting for PS. Specifically, Schöttker et al. [56] examined the association between polypharmacy and non-cancer mortality while focusing on how best to adjust for morbidity as a confounder by indication. This study found that polypharmacy was not associated with non-cancer mortality.

## Correlations

Fig 2 showed that certain groups of cardiovascular medicine (C07-C10) were mainly positively correlated with these same groups. This shows, that older adults with cardiovascular conditions take multiple cardiovascular medications leading to polypharmacy. This is supported by findings from other studies, where cardiovascular medicine has been found to be one of the main contributors to polypharmacy [57]. Rather strong positive correlations were found for the dermatological medications (D01–D11) and gynaecological medications (G01-G02) that largely had positive correlations with all other therapeutic groups. This may reflect that older adults taking these medications, in general, have a very high medication intake, whereby these medications are taken with several other medications.

## Strengths and limitations

This study included all older adults aged 65 or more years registered as living in Denmark at some point during the study period, which limits the risk of selection bias. Furthermore, drawing on information from the registers made it possible to investigate medication use on an individual level and measure polypharmacy monthly, which is more fine-grained than has been possible for most previous studies. Conducting a longitudinal study made it possible to calculate incidence rates of polypharmacy over a five-year period, which makes this study one of few studies to investigate the incidence of polypharmacy. Being able to do this on a nationwide scale with a relatively long follow-up period is a major strength of this study.

We note the following limitations. Firstly, this study investigated polypharmacy as the number of different medications taken per month based on information on prescription drugs from The Danish National Prescription Registry, however, information on hospital provided medication and over-the-counter medication are not accessible in this register. Furthermore, polypharmacy was defined according to purchases of prescription drugs; however, there are no records of whether older adults adhered to drug therapy. Data availability and non-adherence may therefore influence the reliability of the prevalence and incidence estimates. We were able to calculate polypharmacy exposure per month, however, for this purpose we had to use DDD values to calculate the expected duration of intake. Nonetheless, the expected duration may deviate from the actual duration and misclassification may occur as a result. We do not assess the appropriateness of polypharmacy on an individual level, as it would warrant a clinical judgement of each polypharmacy case. Thus, some combinations of medication may not contribute to the risk of premature death. Future studies, should look into which medications are most often combined such as Christensen et al. [6] did when identifying drug profiles and conduct investigations on how these different combinations of medications are associated with adverse drug events including premature death.

Secondly, we may not have adjusted for all potential confounders, as the registers does not contain information on e.g. health behavior and biological parameters, hence, residual confounding may bias the results. Moreover, we adjust for multimorbidity as a binary indicator

based on a list of 47 conditions, which is subject to limitations. The comprehensive list of conditions is a strength, however, the measure neglects to consider the severity of the conditions. Thus, individuals may have the same conditions but be affected by it in very different ways due to different degrees of severity. However, the data we had available from the registers did not hold information on disease severity. Moreover, the simplicity of a binary measure of multimorbidity ignores the degree of multimorbidity. Being multimorbid with two conditions may be very different than being multimorbid with 5 conditions in terms of risk of being exposed to polypharmacy as well as the risk of premature death. Moreover, number of conditions may act as an effect modifier in which the association between polypharmacy and death varies with degree of multimorbidity. This has been found to be the case for hospitalisation as outcome and one could expect the same to be the case for death [58]. Future research should investigate this modification of risk.

## Conclusion

Designed as a nationwide register-based and longitudinal study, this study found that a substantial proportion of older adults were exposed to polypharmacy at some point during their life. Specifically, one third of older adults were exposed to polypharmacy at the time of inclusion, whereas nearly half of the remaining transitioned into polypharmacy during follow-up. Moreover, older adults exposed to polypharmacy or excessive polypharmacy had a substantially increased risk of premature death compared to older adults who were not exposed. These findings support the notion that polypharmacy in and of itself may have adverse health consequences.

Findings from this study shows that polypharmacy is common among older adults, though we do not uncover its appropriateness. Future research should aim to describe the appropriateness of polypharmacy in order for us to gain a deeper understanding of the exposure and its relation to death. On a policy level, it is of interest to reduce exposure to inappropriate polypharmacy. Thus, medication use could be monitored on a national scale, which is the case for various aspects of health. With continuous monitoring of medication intake, trends in medication use over a longer period of time could be observed, which information could be a driver for the development of updated health policies and clinical procedures in order to prevent and reduce polypharmacy in patients where the drug regimen is inappropriate.

## Supporting information

**S1 Table. Algorithms for identification of chronic conditions.**
(DOCX)

**S2 Table. Population characteristics across polypharmacy status before and after applying weights (N = 1,338,058).**
(DOCX)

**S3 Table. Population characteristics across excessive polypharmacy status before and after applying weights (N = 1,338,058).**
(DOCX)

**S4 Table. Percentage of medications taken by those exposed at baseline.**
(DOCX)

**S5 Table. Overview of ATC codes excluded in the sensitivity analysis.**
(DOCX)

**S6 Table. Mean medication use and proportion of polypharmacy cases in the main analysis and sensitivity analysis.**
(DOCX)

## Acknowledgments

The authors wish to thank Helena Egeris Nielsen and Johanne Snog Gillesberg for their contribution in data extraction and data processing.

## Author Contributions

**Conceptualization:** Anna Vera Jørring Pallesen, Rudi G. J. Westendorp, Laust Hvas Mortensen.

**Data curation:** Anna Vera Jørring Pallesen, Laust Hvas Mortensen.

**Formal analysis:** Anna Vera Jørring Pallesen.

**Funding acquisition:** Rudi G. J. Westendorp.

**Investigation:** Anna Vera Jørring Pallesen.

**Methodology:** Anna Vera Jørring Pallesen, Laust Hvas Mortensen.

**Resources:** Laust Hvas Mortensen.

**Validation:** Anna Vera Jørring Pallesen, Laust Hvas Mortensen.

**Visualization:** Anna Vera Jørring Pallesen.

**Writing – original draft:** Anna Vera Jørring Pallesen.

**Writing – review & editing:** Anna Vera Jørring Pallesen, Maria Kristiansen, Rudi G. J. Westendorp, Laust Hvas Mortensen.

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
