## [Decision Letter · Decision Letter 0]

20 Jul 2021

PONE-D-21-15519

Polypharmacy occurrence and the related risk of premature death among older adults in Denmark: A nationwide register-based cohort study

PLOS ONE

Dear Dr. Pallesen,

Thank you for submitting your manuscript to PLOS ONE. After careful consideration, we feel that it has merit but does not fully meet PLOS ONE’s publication criteria as it currently stands. Therefore, we invite you to submit a revised version of the manuscript that addresses the points raised during the review process.

We look forward to receiving your revised manuscript.

Kind regards,

Omid Beiki, M.D., Ph.D.

Academic Editor

PLOS ONE

Journal Requirements:

Reviewers' comments:

Reviewer's Responses to Questions

**Comments to the Author**

1. Is the manuscript technically sound, and do the data support the conclusions?

Reviewer #1: Partly

2. Has the statistical analysis been performed appropriately and rigorously? 

Reviewer #1: Yes

3. Have the authors made all data underlying the findings in their manuscript fully available?

Reviewer #1: No

4. Is the manuscript presented in an intelligible fashion and written in standard English?

Reviewer #1: Yes

5. Review Comments to the Author

Reviewer #1: This paper uses a nationwide register to describe the epidemiology of polypharmacy in Denmark. The paper finds that polypharmacy is common, and associated with several important factors including aging, multimorbidity and socioeconomic factors. It also describes the incidence of polypharmacy, which is poorly reported in the literature, and assesses the association with mortality.

The paper is clearly written and easy to read. The background is well laid out and sets out the aims in a clear and justified manner. The methods seem, in general, appropriate. The results are well structure and easy to follow. The discussion is appropriate, although I think the interpretation needs to be cautious. Data are not available due to the confidential nature of the national dataset used.

I have a number of comments that would benefit from consideration by the authors:

1. Abstract - the final sentence concludes that the strong association with mortality infers a need for moniotoring and policy to *reduce* polypharmacy. This implies a causal association which I do not believe is convincingly supported by the evidence presented.

2. Methods - I am not familiar with the overlap weighting approach used, and I suspect many other readers will not be either. Could the authors expand on this a little, in terms of elaborating on what the benefits of this particular approach are, without the reader needing to resort to reading the references given?

3. Results - The presentation of prevalence over time is, at least initially, confusing - I was left trying to work out why the prevalence drops to 2.4 from a mean of 3.5 in 2013. I was relieved that the discussion concurred with my own conclusions, in that the later years represent younger people entering the study. However, it did make me wonder how useful this is - I would suggest that the analysis focuses on 2013 only in 2013.

4. Results - The presentation of polypharmacy incidence is very nice, given this is not well reported. As is acknowledged by the authors, the 2013 values are less interpretable, and I wonder if tables 3 and 4 should focus on 2014-2017 only, dropping 2013 and changing the reference category accordingly.

5. Discussion: mortality and strengths/limitations sub-sections. The authors state "This strength makes the estimate ...the most reliable ... to describe the association between polypharmacy and mortality." I don't disagree that the association is clearly described, but I do think there are important points that are not considered. Firstly, severity of disease is not accounted for - people may have similar morbidities, but very different degrees of severity of those different morbidities, with severity thus being associated with both mortality and with polypharmacy. Clearly, most studies using routine data cannot readily account for severity as it is not captured in coded records. Nevertheless - this should be acknowledged here and in the strengths/limitations section (at present, the authors state "we may not have adjusted for all potential confounders" - severity is clearly one of these). Secondly, the authors appear to have only measured morbidity using a simply dichotomy (0/1 OR 2+ conditions) based on 47 chronic conditions. Why was there no adjustment for a more sophisticated measure of multimorbidity that accounted for different numbers, types and weightings of conditions? I apologise if this was done, in which case it should be made clearer. Thirdly, the association between polypharmacy and adverse outcome (hospitalisation) has been shown to vary with the degree of multimorbidity (pubmed.ncbi.nlm.nih.gov/24428591/) but no effect modification was considered here; this is worth discussion as one would expect a similar picture to be seen with mortality.

6. The conclusion states that the findings show "a potential for prevention of inappropriate polypharmacy". I disagree - the authors have made no attempt to quantify the appropriateness or otherwise of polypharmacy in the study. The assumption can certainly be made that much of the polypharmacy experienced by this cohort will be inappropriate, but no data is presented to support this argument. Similarly, the conclusion moves on to discuss prevention and reduce of polypharmacy (similar to the abstract) but the data do not support these arguments as a causal association cannot be assumed. The conclusion here, and in the abstract, should be toned down accordingly.

6. PLOS authors have the option to publish the peer review history of their article (what does this mean?). If published, this will include your full peer review and any attached files.

Reviewer #1: **Yes: **Rupert Payne

---

## [Author Response · Author response to Decision Letter 0]

3 Sep 2021

We appreciate the feedback and comments that we have received from the reviewer. In this document we respond to all the comments given and describe the changes that we have made to the manuscript. We hope that these modifications will satisfy the reviewer. 

Comment 1: Abstract - the final sentence concludes that the strong association with mortality infers a need for monitoring and policy to *reduce* polypharmacy. This implies a causal association which I do not believe is convincingly supported by the evidence presented.

Response 1: Thank you for this feedback, that is a very good point. We understand that on the basis of our study, we cannot draw conclusions on causality. We have edited the abstract accordingly and hopefully to your satisfaction (Line 43-45).

Comment 2: Methods - I am not familiar with the overlap weighting approach used, and I suspect many other readers will not be either. Could the authors expand on this a little, in terms of elaborating on what the benefits of this particular approach are, without the reader needing to resort to reading the references given?

Response 2: The overlap weighting approach is a relatively new approach – or at least it is not widely used yet. So in that regard, we agree that the description of this method could be expanded. The paragraph on this topic (p. 8, lines 153-159) has been edited according to your feedback. 

Comment 3: Results - The presentation of prevalence over time is, at least initially, confusing - I was left trying to work out why the prevalence drops to 2.4 from a mean of 3.5 in 2013. I was relieved that the discussion concurred with my own conclusions, in that the later years represent younger people entering the study. However, it did make me wonder how useful this is - I would suggest that the analysis focuses on 2013 only in 2013.

Response 3: By design, we look at prevalence at the time of inclusion and as we study polypharmacy in an open cohort, the age structure in the first year will be different than in the subsequent years. All people older than 65 years will enter in 2013, while only 65 year-olds will enter in the later years. To accommodate your point that it might be a bit confusing, we have added a note to the table (p. 14). Additionally, we have added an explanation in the results section (p. 11, lines 190-199) where we explain the difference in age structure and also share results from a descriptive analysis of the mean medication use and prevalence among the 65 year-olds that were included in 2013. This analysis reveal that the mean medication use were 2.4, while 18.6% had polypharmacy and 2.5% had excessive polypharmacy at the time of inclusion. This additional analysis therefore shows that the 65 year-olds in 2013 do not differ from the 65 year-olds included in 2014-2017. We hope that this addition will be satisfactory. 

Comment 4: Results - The presentation of polypharmacy incidence is very nice, given this is not well reported. As is acknowledged by the authors, the 2013 values are less interpretable, and I wonder if tables 3 and 4 should focus on 2014-2017 only, dropping 2013 and changing the reference category accordingly.

Response 4: Thank you for this comment, we are happy that we can report on the incidence of polypharmacy since – as you say – this is not well reported. 

We investigate incidence among people who were not exposed at the time of inclusion. Due to differences in the age structure as described in the previous response, individuals entering the study population in 2013 differs from individuals entering the study population in any of following years in terms of age. So by design, the differences in incidence rates observed in 2013 compared to the subsequent years are expected. Similar to our response to your previous comment, we have decided to make a note of this difference in Tables 3 and 4 on page 17 and 19. Moreover, to limit the effect of different age structures across the years we adjusted for age in the Cox regression model II (Table 3 and 4). The results show that when adjusting for age as well as sex and multimorbidity, the hazard of polypharmacy is not significantly different across the years of inclusion – only 2017 shows an increased hazard of transitioning into polypharmacy. We have emphasized this on page 25, line 324-327. 

Comment 5: Discussion: mortality and strengths/limitations sub-sections. The authors state "This strength makes the estimate ...the most reliable ... to describe the association between polypharmacy and mortality." I don't disagree that the association is clearly described, but I do think there are important points that are not considered. Firstly, severity of disease is not accounted for - people may have similar morbidities, but very different degrees of severity of those different morbidities, with severity thus being associated with both mortality and with polypharmacy. Clearly, most studies using routine data cannot readily account for severity as it is not captured in coded records. Nevertheless - this should be acknowledged here and in the strengths/limitations section (at present, the authors state "we may not have adjusted for all potential confounders" - severity is clearly one of these). Secondly, the authors appear to have only measured morbidity using a simply dichotomy (0/1 OR 2+ conditions) based on 47 chronic conditions. Why was there no adjustment for a more sophisticated measure of multimorbidity that accounted for different numbers, types and weightings of conditions? I apologies if this was done, in which case it should be made clearer. Thirdly, the association between polypharmacy and adverse outcome (hospitalisation) has been shown to vary with the degree of multimorbidity (pubmed.ncbi.nlm.nih.gov/24428591/) but no effect modification was considered here; this is worth discussion as one would expect a similar picture to be seen with mortality.

Response 5: These are some very valuable considerations for the discussion of methods. 

We do agree with your argument that the severity of the morbidities are important in terms of the risk of being exposed to polypharmacy as well as the risk of dying. The severity of the disease constitute an indication for both. Our study is based on register data, which – as you also mention – unfortunately do not hold information about the severity of the disease. Nonetheless, it is an important point, which we have highlighted in the discussion line 415-419. 

Regarding your second point on our use of a binary morbidity variable based on 47 chronic conditions, we chose this purposefully. We were interesting in the exposure to multimorbidity defined as having at least two conditions at the same time. All of the conditions listed most often require drug therapy. And some of the conditions require therapy with multiple drugs. Intuitively, the risk of polypharmacy should be higher among those with multiple conditions. We actually did conduct the analysis with multimorbidity categorized as 0, 1, 2, 3, 4, and 5+. Unsurprisingly, the risk/hazard of being exposed to polypharmacy increased with a larger number of conditions. However, the interpretation of the results were difficult due to very high OR and HR. We decided that a simple binary measure of multimorbidity were better for interpretation. However, we do acknowledge that such a simplification has its limitations. We have elaborated on this in the discussion of strengths/weaknesses in line 419-422. 

In regards to your last comment, you make a very interesting point that multimorbidity may act as an effect modifier. We propose in the paper, that this is investigated in future research. Please see line 422-426.

Comment 6: The conclusion states that the findings show "a potential for prevention of inappropriate polypharmacy". I disagree - the authors have made no attempt to quantify the appropriateness or otherwise of polypharmacy in the study. The assumption can certainly be made that much of the polypharmacy experienced by this cohort will be inappropriate, but no data is presented to support this argument. Similarly, the conclusion moves on to discuss prevention and reduce of polypharmacy (similar to the abstract) but the data do not support these arguments as a causal association cannot be assumed. The conclusion here, and in the abstract, should be toned down accordingly.

Response 6: As for your first comment regarding our abstract, we do agree that make too strong conclusions based on the study that we conducted. We are fully aware that we can draw no conclusion on causality based on our study. Therefore, we have edited the conclusion accordingly and toned it down (lines 436-440).

---

## [Decision Letter · Decision Letter 1]

11 Jan 2022

PONE-D-21-15519R1Polypharmacy occurrence and the related risk of premature death among older adults in Denmark: A nationwide register-based cohort studyPLOS ONE

Dear Dr. Pallesen,

Thank you for submitting your manuscript to PLOS ONE. After careful consideration, we feel that it has merit but does not fully meet PLOS ONE’s publication criteria as it currently stands. Therefore, we invite you to submit a revised version of the manuscript that addresses the points raised during the review process.

We look forward to receiving your revised manuscript.

Kind regards,

Omid Beiki, M.D., Ph.D.

Academic Editor

PLOS ONE

Journal Requirements:

Reviewers' comments:

Reviewer's Responses to Questions

**Comments to the Author**

1. If the authors have adequately addressed your comments raised in a previous round of review and you feel that this manuscript is now acceptable for publication, you may indicate that here to bypass the “Comments to the Author” section, enter your conflict of interest statement in the “Confidential to Editor” section, and submit your "Accept" recommendation.

Reviewer #1: (No Response)

2. Is the manuscript technically sound, and do the data support the conclusions?

Reviewer #1: Yes

3. Has the statistical analysis been performed appropriately and rigorously? 

Reviewer #1: Yes

4. Have the authors made all data underlying the findings in their manuscript fully available?

Reviewer #1: No

5. Is the manuscript presented in an intelligible fashion and written in standard English?

Reviewer #1: Yes

6. Review Comments to the Author

Reviewer #1: Thank you to the authors for addressing the points I raised. Although I still have some reservations over the manner in which the results are presented, I feel the clarifications that have been made help address these concerns.

My only outstanding (minor) remark is the response to my comment 2 (methods). The response was generally fine, but the statement "limitations include that large weights are given for individual participants, large variability in the estimated treatment effect may be observed and bias" seems to be incomplete, although it may be simply a grammatical issue. I think it perhaps should read "limitations include that large weights are given for individual participants, large variability in the estimated treatment effect may be observed, and THERE MAY BE bias"? (if so, bias in what sense?) Or alternatively was the term "and bias" supposed to be deleted and the word "and" added prior to "although"?

7. PLOS authors have the option to publish the peer review history of their article (what does this mean?). If published, this will include your full peer review and any attached files.

Reviewer #1: **Yes: **Rupert Payne

---

## [Author Response · Author response to Decision Letter 1]

7 Feb 2022

Thank you so much for your positive feedback on the revised manuscript. Your suggestions have definitely helped us improve it. 

The one comment you give to our revised manuscript surely warrants some attention. We acknowledge that the word bias cannot be added without further explanation. We therefore propose another sentence to replace the one in question (p. 8, lines 155-158).

---

## [Editor Report · Decision Letter 2]

9 Feb 2022

Polypharmacy occurrence and the related risk of premature death among older adults in Denmark: A nationwide register-based cohort study

PONE-D-21-15519R2

Dear Dr. Pallesen,

We’re pleased to inform you that your manuscript has been judged scientifically suitable for publication and will be formally accepted for publication once it meets all outstanding technical requirements.

Kind regards,

Omid Beiki, M.D., Ph.D.

Academic Editor

PLOS ONE
---

## [Editor Report · Acceptance letter]

11 Feb 2022

PONE-D-21-15519R2 

Polypharmacy occurrence and the related risk of premature death among older adults in Denmark: A nationwide register-based cohort study 

Dear Dr. Pallesen:

I'm pleased to inform you that your manuscript has been deemed suitable for publication in PLOS ONE. Congratulations! Your manuscript is now with our production department. 

Kind regards, 

on behalf of

Dr. Omid Beiki 

Academic Editor

PLOS ONE